# Engines of change: Transposable element mutation rates are high and variable within *Daphnia magna*

Eddie K. H. Ho[1][☉], Emily S. Bellis[1,2][☉], Jaclyn Calkins[1,3], Jeffrey R. Adrion[4], Leigh C. Latta IV[1,5], Sarah Schaack[1]*

1 Department of Biology, Reed College, Portland, Oregon, United States of America, 2 Department of Computer Science, Arkansas State University, Jonesboro, Arkansas, United States of America, 3 College of Human Medicine, Michigan State University, East Lansing, Michigan, United States of America, 4 Institute of Ecology and Evolution, University of Oregon, Eugene, Oregon, United States of America, 5 Lewis-Clark State College, Lewiston, Idaho, United States of America

☉ These authors contributed equally to this work.
* schaack@reed.edu

**Data Availability Statement:** WGS data have been deposited at NCBI (PRJNA658680) and all code is available online (https://github.com/EddieKHHo/

## Abstract

Transposable elements (TEs) represent a major portion of most eukaryotic genomes, yet little is known about their mutation rates or how their activity is shaped by other evolutionary forces. Here, we compare short- and long-term patterns of genome-wide mutation accumulation (MA) of TEs among 9 genotypes from three populations of *Daphnia magna* from across a latitudinal gradient. While the overall proportion of the genome comprised of TEs is highly similar among genotypes from Finland, Germany, and Israel, populations are distinguishable based on patterns of insertion site polymorphism. Our direct rate estimates indicate TE movement is highly variable (net rates ranging from -11.98 to 12.79 x $10^{-5}$ per copy per generation among genotypes), differing both among populations and TE families. Although gains outnumber losses when selection is minimized, both types of events appear to be highly deleterious based on their low frequency in control lines where propagation is not limited to random, single-progeny descent. With rate estimates 4 orders of magnitude higher than base substitutions, TEs clearly represent a highly mutagenic force in the genome. Quantifying patterns of intra- and interspecific variation in TE mobility with and without selection provides insight into a powerful mechanism generating genetic variation in the genome.

## Author summary

Transposable elements (TEs) are a significant portion of most eukaryotic genomes, yet our understanding of their rates of mobility and their patterns of accumulation remain very limited. Here, we estimate genome-wide rates of gain and loss of TEs in *Daphnia magna*, a well-studied model organism in ecology, and compare these rates of mutation to the long-term accumulation of TEs in the genome. Rates vary remarkably among genotypes and populations, and between different types of TEs within the same lineage.

DaphiaMagna_MA_TE). The TE library used is available as S1 Data.

**Funding:** This work was supported by awards from the National Institute of General Medical Sciences of the National Institutes of Health (GM132861) and National Science Foundation (MCB-1150213) to SS. The funders had no role in study design, data collection and analysis, decision to publish, or preparation of the manuscript.

**Competing interests:** The authors have declared that no competing interests exist.

Despite this variation, over long time periods, TE content in the genome is extremely similar across genotypes within the species. We compare our results to the few estimates available from other taxa, and argue that TEs are an important source of mutagenesis in the genome worthy of further investigation.

## Introduction

It is now known that transposable elements (TEs) make up a significant proportion of the genome in most eukaryotes, and in some cases even represent the majority of the sequence (e.g., [1–3]). Although commonly referred to as 'junk DNA' or genomic 'parasites', and therefore masked (or removed) in genomic analyses in favor of focusing on genic regions [4], the importance of TEs is gaining wider appreciation and the repetitive landscape of the genome is no longer ignored [5, 6]. Notably, there are now many high profile examples of TEs performing functional roles in the host genome (e.g., [7]) and recent work has cited their role in numerous biological processes, such as adaptation and speciation (e.g., [8–10]). The potential influence of TEs at the genomic level, whether structural (e.g., [11]), direct (e.g., contributing new coding or regulatory sequences; [12]), or indirect (e.g., changing the epigenomic landscape of the host genome; [13, 14]), is now known to be significant [15].

Because TEs are mobile and far outnumber 'regular' protein-coding genes in most eukaryotic genomes, elucidating their patterns of replication, transposition, and excision/deletion is a major task that spans subdisciplines from molecular biology to population genetics [16, 17]. Understanding the dynamics of TE proliferation includes knowing how TEs jump between lineages (horizontal transfer of TEs [HTT]; [18, 19]), differential success among TE families in various host lineages (e.g., [20]), and how TEs "die" or go extinct, or are resurrected (e.g., [21]). Indeed, the idea that genomes are like habitats and that TEs are like individuals (and TE families like species) has gained popularity as a way of characterizing the complexities of TE activity in different host genomes (e.g., [22]). Furthermore, the notion that TEs and their host genomes co-evolve is now widely acknowledged [23]. On average, the effects of new TE insertions, like all spontaneous mutations, are thought to be deleterious, although there are long-standing debates about whether the majority of these negative effects are direct (e.g., interrupting genes) or indirect (e.g., increasing the risk of ectopic recombination) [24]. More broadly, the outcomes of TE activity in host genomes is increasingly a target of investigation and is known to range from beneficial to neutral to deleterious [25].

Ultimately, the TE content observed in a lineage is the net product of the intrinsic mutational properties of the TEs, combined with the host genome's cellular and genomic defense system, which is then acted upon (over evolutionary time scales) by population genetic factors such as the strength of selection and genetic drift. An important question is to what degree the genetic variation generated by TEs is altered or retained in natural populations. If selection can operate efficiently, TEs should not accumulate to high copy number, unless their mutation rates are very high. On the other hand, if effective population sizes or recombination rates are low, selection may not act efficiently, and TEs could accumulate even with low rates of gain [26, 27]. Comparing TE dynamics in the laboratory versus in natural populations can reveal the relative roles of mutation, selection, and drift. Furthermore, quantifying TE dynamics among closely-related lineages reveals how the mutational process and/or evolutionary constraints vary within and between genotypes, populations, and species due to host differences. Finally, contrasting the rate and spectra of TE mutations with other types of more well-studied mutations (e.g., base substitutions) or mutational processes that might affect their spread (e.g., gene conversion) is critical for understanding how, and how fast, genetic variation is generated.

Here, we compare patterns of TE activity over short time periods using a mutation accumulation (MA) experiment, where selection is minimized, to patterns of long-term accumulation by comparing TE content among genotypes from multiple populations and between congeners using *Daphnia*. *Daphnia* are an excellent model organism for studying TEs (and mutations, more broadly) because they can reproduce asexually, removing the complicating influence of meiosis and sex on proliferation, and have been shown to have high mutation rates for other categories of mutation [28–30]. *Daphnia* are aquatic microcrustaceans (Order: Cladocera) often used in ecological and toxicological studies, but which have more recently become the focus of evolutionary and genomic research [31]. In this study, we quantify the TE profiles of 9 starting genotypes sampled from three populations of *D. magna* across a latitudinal gradient (Finland, Germany, and Israel; S1 Table). We use those same genotypes to perform a multi-year MA experiment to directly estimate rates of gains and losses for all known TEs. We also compare our results to the congener, *D. pulex*, for which some similar data are available, and to mutation rates for other types of mutation that have been measured in *D. magna* previously [29]. While both *D. magna* and *D. pulex* appear extremely similar in morphology, physiology, behavior, distribution, and life-history, they do differ in genome size (*D. magna* > *D. pulex* by ~30%; [32, 33]) and mutation rate (*D. magna* > *D. pulex*; [28, 29, 34]).

Patterns of long-term TE accumulation can be measured in several ways: abundance and diversity of TEs present in the genome, insertion site polymorphism among lineages, and mean pairwise divergences (MPDs) of copies of each TE family, where lower values are assumed to represent more recent activity because copies will not have diverged yet due to the accumulation of point mutations. Direct observations of TE movement in real-time using MA experiments represent the gold standard for accurate rate estimates, but have been rarely used to quantify rates of TE movement (reviewed in [35]). In MA experiments, descendent lineages are propagated via single-progeny descent from a known ancestor to minimize natural selection and lines are sequenced to count the number of events per copy per generation and calculate rates. Importantly, while there are two kinds of events that can be scored for a particular TE copy—gains and losses—there are a number of ways by which these two events can occur, even in asexually-reproducing lineages. New TE copies can result from insertions (transposition or retrotransposition), duplication events, polyploidization, DNA repair, gene conversion events, and/or ectopic recombination. Similarly, loss of a TE can be due to excision (although not all elements are capable of excision [e.g., Class 1 retroelements]), deletions (if a TE was present in a deleted region), gene conversion, or ectopic recombination events. In the vast majority of cases, the exact mechanism of gain or loss is not known, nor is the degree to which the host genome has co-evolved molecular mechanisms to suppress even active TE families. Furthermore, the likelihood of gain and loss via these different mechanisms may vary, for example among sites that are initially unoccupied, heterozygous, or homozygous for a TE (Fig 1), and thus we predict rates of TE activity to vary among TE families based on a number of factors (including TE type, mechanisms of mobility, copy number, and/or the time since the TE first entered the host genome). Ultimately, our goals are to measure TE mutation rates and determine to what extent they vary across lineages, compare TE dynamics over the short (with and without selection) and long time scales, and determine if rates of TE movement correlate with more frequently measured mutation rates, such as base substitution mutation rates.

## Results

To quantify the long-term patterns of TE accumulation, we surveyed the whole genome of 9 genotypes of *D. magna* from three populations and characterized the TE content using three metrics: 1) overall abundance and diversity, 2) insertion site polymorphism, and 3) mean pairwise

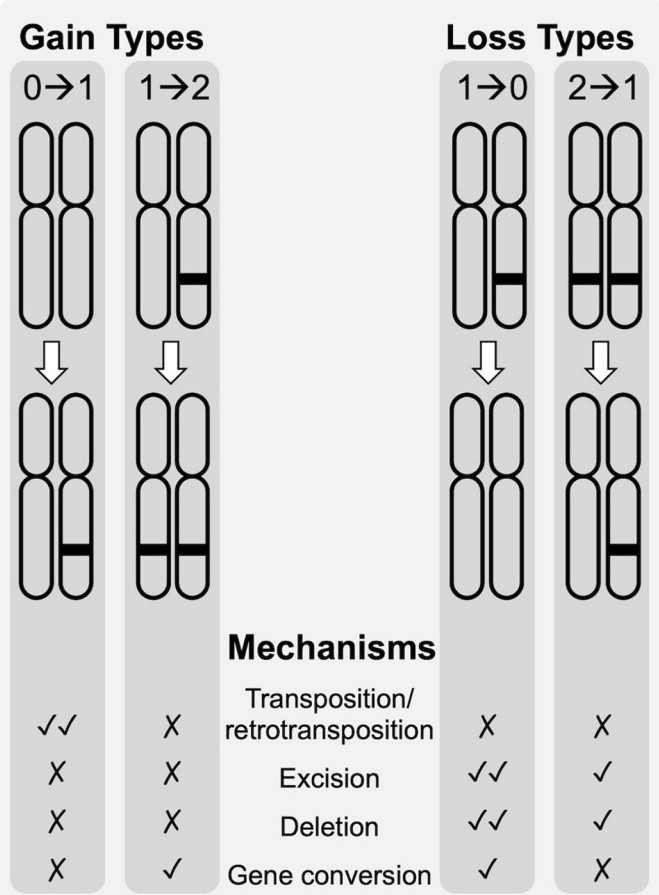

**Fig 1. Categories of loss and gain for TE copies.** Different mechanisms can explain the categories of loss and gain of TEs at a given locus (0→1, 1→2, 1→0, and 2→1) that occur in asexually-reproducing, diploid organisms like *Daphnia magna*. For each type of gain or loss, check marks indicate the qualitative, relative likelihood of a given mechanism and X marks indicate a particular mechanism cannot produce that type of gain or loss.

divergence among copies in each family or superfamily. To quantify short-term patterns of mobility, we directly estimated TE mutation rates (gains and losses; Fig 1) based on events observed during a multi-year mutation accumulation (MA) experiment initiated from each of the same 9 genotypes. In these experiments, descendant lines are either propagated via single-progeny descent (to minimize selection) or maintained at large population sizes (selection is not minimized). We examine intra- and interspecific variation by comparing our results from *D. magna* collected from populations along a latitudinal gradient (Finland, Germany, and Israel; Fig 2A) to the congener, *D. pulex*, wherever possible. *D. magna* and *D. pulex* assemblies were of similarly good quality, possessing N50 of approximately 1 Mb and containing greater than 77% of the complete genes from the Arthropod reference gene set (S2 Table). Lastly, we compare TE mutation rates from *D. magna* to base substitution mutation and gene conversion rates estimated in the same lineages to see if patterns of TE rate variation covary with other mutational processes.

## Characterizing TE content in *Daphnia*

The relative abundance of TEs across the nine *D. magna* genotypes is similar (Fig 2B and Tables 1 and S3 and S4). In *Daphnia*, LTR retrotransposons are the most common type of TEs,

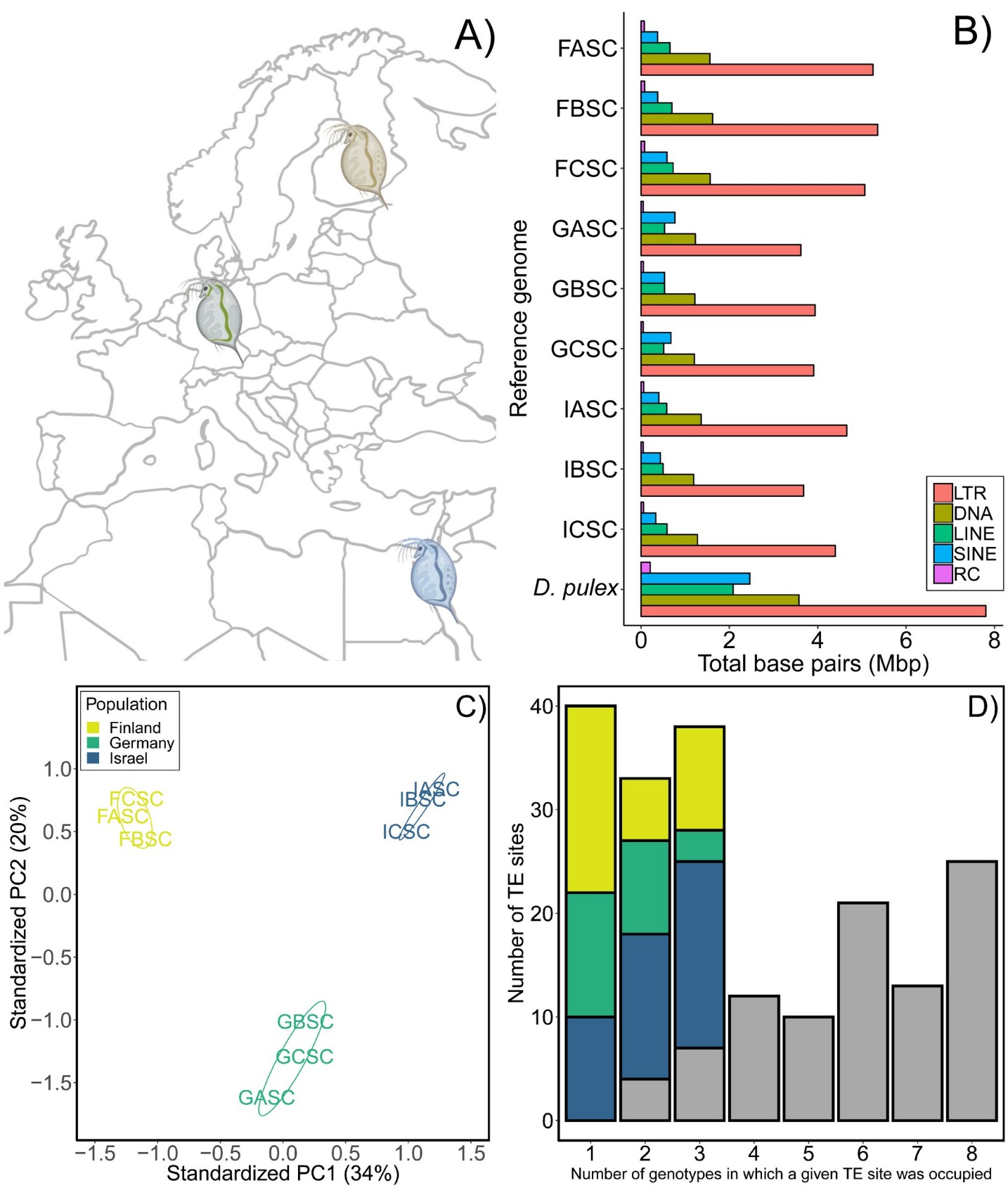

**Fig 2. TE profiles for the 9 starting genotypes of *Daphnia magna*.** (A) Map of the three populations (Finland [FASC, FBSC, FCSC], Germany [GASC, GBSC, GCSC], and Israel [IASC, IBSC, ICSC]) from which genotypes were collected (created with BioRender.com). (B) Abundance and diversity (in millions of bp [Mbp]) per type of TE (Long Terminal Repeats [LTR], DNA transposons [DNA], Long Interspersed Nuclear Elements [LINE], Short Interspersed Nuclear Elements [SINE], and Rolling Circle elements [RC]) compared to *D. pulex* (reference genome; PA42 [BioProject: PRJEB14656]). (C) Principal Component Analysis based on TE insertion polymorphism (TIP) data distinguishes populations based on the presence/absence of TEs (n = 192 polymorphic sites). (D) Number of polymorphic TE sites occupied; the left bar (x = 1) is the number of singletons (sites occupied in only one genotype), colored portions of bars in x = 2 and x = 3 represent sites occupied in 2 and 3 genotypes, respectively, when from the same population. Grey portions of each bar represent the number of sites that were occupied in ≥2 genotypes that were not population-specific. FASC was used as the reference assembly for (C) and (D); see S2 Text.

with the *Gypsy* superfamily being the most abundant (Table 1). All other categories of TEs (DNA transposons, Long and Short Interspersed Nuclear Elements [LINEs and SINEs], and rolling circle [RCs]) constitute less than 2% of the genome, although DNA transposons are still highly diverse with 18 different families represented (S5 Table). Although abundance is consistent within *D. magna* when comparing across genotypes, overall abundance and the abundance of individual families differed between *D. magna* and its congener, *D. pulex* (*D. pulex* > *D. magna*; $t_8$ = -14.2, P < 0.0001; Fig 2B and S6 Table), with 7 and 19 families of DNA transposons being specific to *D. magna* and *D. pulex*, respectively.

It is important to note, TE abundance can be measured in two ways—repeat masking a genome assembly with a TE library or mapping short reads to a TE library to use depth of coverage as an estimate of abundance. In fact, estimates of overall TE content in *D. magna* differ by more than a factor of two using these two methods (6% using repeat masking and 16% using read mapping; S4 Table), which is likely because repeat masking is more sensitive to the quality of the assembly. We recommend using a read-mapping approach for accuracy, however repeat masking is more common and provides the opportunity for inter- as well as intraspecific comparisons here; see S2 Text, S2 Text, and S5 Table for estimates using both methods.

## Variation in TE activity over long time periods

Despite the consistency in terms of abundance, we quantified TE insertion polymorphisms (TIPs) among the 9 genotypes of *D. magna* sampled and were able to clearly distinguish genotypes based on their population-of-origin using principal components analysis (PCA; Fig 2C), regardless of reference genome used (S1 Fig). Depending on the assembly used as reference (see S2 Text), we identified between 1442 and 1903 TE sites, of which 13% to 16% were polymorphic across the 9 genotypes (S7 Table) and which, by k-means clustering, always revealed non-overlapping clusters corresponding to their population-of-origin (S8 Table). On average, we find 19% of TIPs are specific to a single genotype (i.e., singletons) and an additional 29% of TIPs are specific to a single population (Figs 2D and S2 and S3, and S9 and S10 Tables). Whether a particular position in the genome is occupied by a TE is determined by events at multiple levels: the chromosome level (e.g., gains/losses due to insertions, deletions, or gene conversion events,) and/or at the individual/population level (e.g., frequency of sexual reproduction or the strength of selection against new insertions). An additional interpretation of an excess of singletons is that the TE family is, or has recently been, active.

Another indicator of recent activity is low levels of mean pairwise divergence (MPD) among copies belonging to a given TE family because new copies have not yet accumulated point mutations. The range of MPDs across TE families was 15–31%, with SINE elements having the lowest values (S4–S15 Figs). Surprisingly, we observed higher MPDs in TE families that were currently active in our MA experiments (~21% for active families and ~19% for inactive; S2 Text and S11 Table). An alternative explanation for high MPDs is a higher base substitution mutation rate, which has been reported for *D. magna* (greater than *D. pulex*; [29]). While we

**Table 1. Abundance of TE types by family or superfamily for *Daphnia magna* (averaged across nine genotypes) and *D. pulex* (PA42 [PRJNA307976]).**

| TE Type | Family or Superfamily | Percent of assembly | | Active in *D. magna* MA lines? |
|---|---|---|---|---|
| | | *D. magna* | *D. pulex* | |
| DNA | *Academ-1* | 0.05 | 0.01 | Y |
| | *CMC-EnSpm* | 0.15 | 0.23 | Y |
| | *Dada* | 0.00 | 0.08 | N |
| | *hAT* | 0.02 | 0.00 | N |
| | *hAT-Ac* | 0.38 | 0.29 | Y |
| | *hAT-Charlie* | 0.01 | 0.00 | N |
| | *hAT-hATm* | 0.04 | 0.03 | N |
| | *hAT-Tip100* | 0.03 | 0.00 | N |
| | *IS3EU* | 0.00 | 0.05 | N |
| | *Kolobok-H* | 0.00 | 0.07 | N |
| | *Merlin* | 0.06 | 0.00 | N |
| | *MULE* | 0.00 | 0.02 | N |
| | *MULE-F* | 0.00 | 0.05 | N |
| | *MULE-MuDR* | 0.03 | 0.15 | N |
| | *P* | 0.08 | 0.09 | Y |
| | *P-Fungi* | 0.04 | 0.00 | N |
| | *PIF-Harbinger* | 0.05 | 0.05 | N |
| | *PIF-ISL2EU* | 0.04 | 0.03 | Y |
| | *PiggyBac* | 0.00 | 0.05 | N |
| | *Sola-1* | 0.00 | 0.11 | N |
| | *Sola-2* | 0.02 | 0.03 | N |
| | *Sola-3* | 0.00 | 0.06 | N |
| | *TcMar-Fot1* | 0.04 | 0.08 | N |
| | *TcMar-Tc1* | 0.07 | 0.02 | N |
| | *TcMar-Tigger* | 0.00 | 0.05 | N |
| | *Zator* | 0.00 | 0.01 | N |
| | *Zisupton* | 0.04 | 0.00 | N |
| | Unclassified | 0.01 | 0.35 | N |
| | Total | 1.16 | 1.89 | |
| LINE | *I* | 0.22 | 0.05 | Y |
| | *I-Jockey* | 0.03 | 0.05 | N |
| | *L1* | 0.00 | 0.03 | N |
| | *L1-Tx1* | 0.11 | 0.21 | N |
| | *L2* | 0.00 | 0.30 | N |
| | *Penelope* | 0.02 | 0.09 | Y |
| | *R1* | 0.02 | 0.11 | N |
| | *R1-LOA* | 0.00 | 0.05 | N |
| | *R2-NeSL* | 0.10 | 0.16 | N |
| | *Rex-Babar* | 0.00 | 0.02 | N |
| | *Tad1* | 0.00 | 0.04 | N |
| | Total | 0.51 | 1.10 | |
| LTR | *Copia* | 0.35 | 0.69 | N |
| | *DIRS* | 0.25 | 0.28 | Y |
| | *ERV1* | 0.00 | 0.04 | N |
| | *ERVK* | 0.00 | 0.10 | N |
| | *Gypsy* | 2.12 | 1.84 | Y |

*(Continued)*

**Table 1.** (Continued)

| TE Type | Family or Superfamily | Percent of assembly | | Active in *D. magna* MA lines? |
|---|---|---|---|---|
| | | *D. magna* | *D. pulex* | |
| | *Ngaro* | 0.08 | 0.04 | N |
| | *Pao* | 0.94 | 1.12 | Y |
| | Unclassified | 0.05 | 0.02 | N |
| | Total | 3.79 | 4.12 | |
| SINE | *5S-Deu-L2* | 0.00 | 0.14 | N |
| | *ID* | 0.02 | 0.02 | N |
| | *tRNA-Core-RTE* | 0.00 | 0.05 | N |
| | *tRNA-V-CR1* | 0.02 | 0.00 | N |
| | Unclassified | 0.40 | 1.09 | N |
| | Total | 0.43 | 1.30 | |
| RC | *Helitron* | 0.05 | 0.11 | N |
| Retroposon | *L1-dep* | 0.00 | 0.03 | N |
| TOTAL | | 5.94 | 8.82 | |

*Abundance estimates based on RepeatMasker method (see Methods for details).

observed interspecific differences in MPDs across TE families between the two species, they were not consistently higher in *D. magna* as one would predict (S12 Table), nor did they correlate with known intraspecific variation in base substitution mutation rates within this species (S16 Fig; $\rho$ = -0.66, $t_7$ = -2.3, P = 0.055).

### Estimated rates of TE loss and gain using mutation accumulation experiments

We used mutation accumulation (MA) experiments initiated from each of the 9 genotypes of *D. magna* from each of the three populations to estimate overall (Tables 2 and S13) and family-specific TE mutation rates (Table 3). Using whole genome sequence (WGS) data from the MA lines, we detected 67 gain and 28 loss mutations; S14 Table shows the location and read support for each event. Rates of gain across MA lines ranged from 0 to 22.6 x $10^{-5}$ per copy per generation with a mean rate of 1.39 x $10^{-5}$ /copy/gen (95% CI: 0.41 x $10^{-5}$–2.66 x $10^{-5}$) and loss rates ranged

**Table 2. Number of events and mean TE mutation rates (per copy per generation, including 95% confidence intervals [CI]) for gains and losses based on whole genome sequence data from 66 *Daphnia magna* mutation accumulation and extant control lines descended from 9 starting genotypes collected from Finland, Germany, and Israel.**

| | Mutation Accumulation Lines | | | | Extant Control Lines | | | |
|---|---|---|---|---|---|---|---|---|
| | Mutation rate ($\times 10^{-5}$) per copy/ per generation | | | | Mutation rate ($\times 10^{-5}$) per copy/ per generation | | | |
| | Number of events | Mean | Lower CI | Upper CI | Number of events | Mean | Lower CI | Upper CI |
| Gain (all kinds) | 67 | 1.39 | 0.41 | 2.66 | 2 | 0.002 | 0 | 0.005 |
| *0-->1 gain* | 62 | 1.17 | 0.23 | 2.42 | 1 | 0.001 | 0 | 0.002 |
| *1-->2 gain* | 5 | 0.22 | 0 | 0.63 | 1 | 0.001 | 0.000 | 0.004 |
| Loss (all kinds) | 28 | 1.7 | 0.53 | 3.23 | 17 | 0.23 | 0.064 | 0.46 |
| *2-->1 loss* | 2 | 0.04 | 0 | 0.09 | 9 | 0.11 | 0.02945 | 0.21 |
| *1-->0 loss* | 26 | 1.67 | 0.46 | 3.21 | 8 | 0.12 | 0.004 | 0.33 |
| Total | 95 | 3.09 | 1.37 | 5.14 | 19 | 0.23 | 0.064 | 0.47 |

*Confidence intervals estimated by bootstrapping across MA lines 10000 times.

**Table 3. Number of events and rates (plus 95% confidence intervals [CI]) of gain and loss (per copy per generation) for each TE superfamily in which events were observed averaged across all MA lines.** Gains and losses based on whole genome sequence data from *Daphnia magna* mutation accumulation lines descended from 9 starting genotypes collected from Finland, Germany, and Israel.

| Type | Family/ Superfamily | Gains | | | | Losses | | | |
|---|---|---|---|---|---|---|---|---|---|
| | | Number of events | Mean Rate ($\times 10^{-5}$) | Lower CI | Upper CI | Number of events | Mean Rate ($\times 10^{-5}$) | Lower CI | Upper CI |
| DNA | *Academ-1* | 1 | 4.9 | 0.0 | 14.6 | 0 | - | - | - |
| | *CMC-EnSpm* | 0 | - | - | - | 1 | 0.3 | 0.0 | 1.0 |
| | *hAT-Ac* | 1 | 3.6 | 0.0 | 10.9 | 3 | 5.9 | 0.0 | 14.7 |
| | *P* | 0 | - | - | - | 1 | 9.0 | 0.0 | 27.1 |
| | *PIF-ISL2EU* | 1 | 10.1 | 0.0 | 30.3 | 1 | 3.3 | 0.0 | 10.0 |
| LINE | *I* | 0 | - | - | - | 1 | 4.4 | 0.0 | 13.1 |
| | *Penelope* | 1 | 11.0 | 0.0 | 32.9 | 0 | - | - | - |
| LTR | *DIRS* | 0 | - | - | - | 2 | 10.2 | 0.0 | 27.2 |
| | *Gypsy* | 59 | 7.9 | 3.5 | 13.4 | 11 | 6.1 | 1.0 | 13.1 |
| | *Pao* | 4 | 0.8 | 0.1 | 1.8 | 8 | 7.2 | 1.2 | 15.6 |

*Confidence intervals estimated by bootstrapping across MA lines 10000 times.

from 0 to 31.8 x $10^{-5}$ /copy/gen with a mean of 1.70 x $10^{-5}$ /copy/gen (95% CI: 0.53 x $10^{-5}$–3.23 x $10^{-5}$; S15 Table). Looking across genotypes, averaging across rates for all TE families (with non-zero copy numbers in all genotypes), it is clear that some genotypes have a bias towards gains while others exhibit mainly losses (Fig 3A and S16 Table). To test for a population effect, we fit a

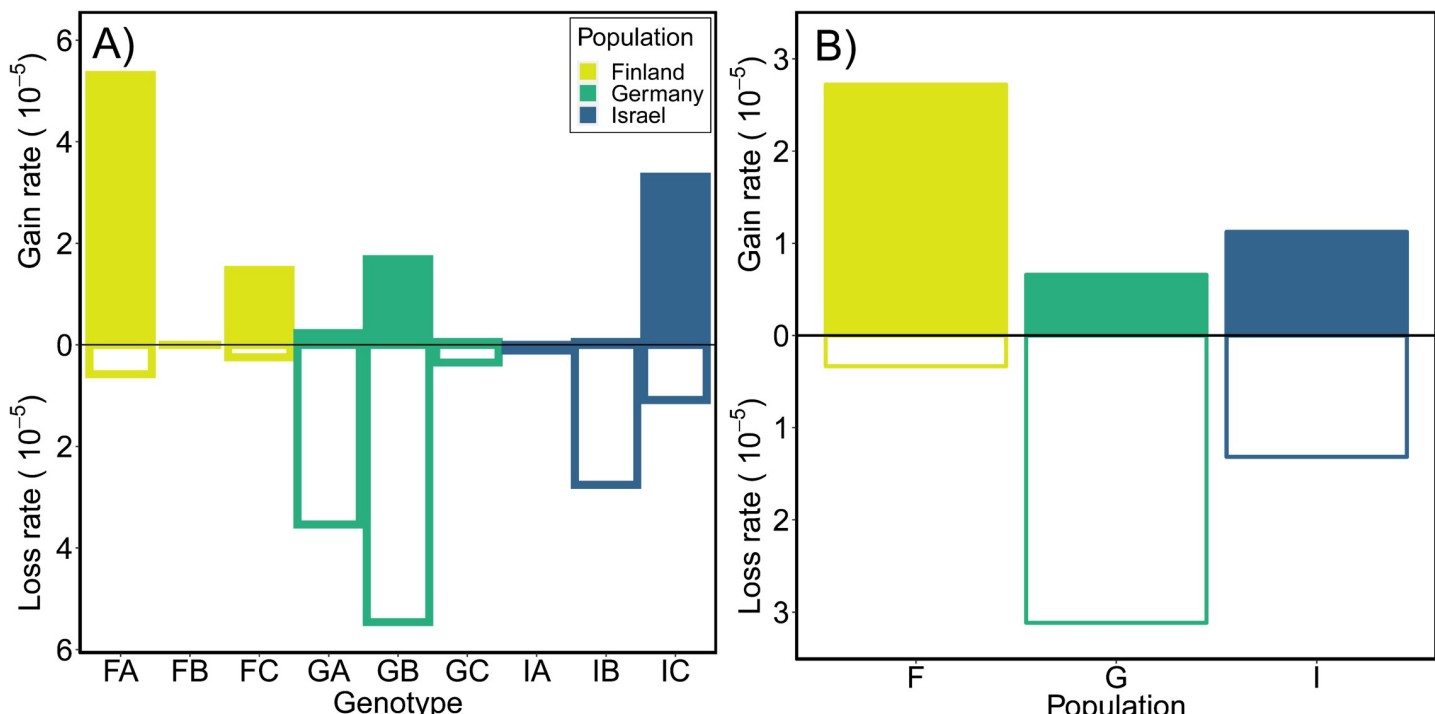

**Fig 3. TE gain and loss rates for *D. magna* MA lines.** Gain and loss rates (per copy per generation) for each (A) genotype and (B) population of *D. magna* averaged across all TE families. Mean rates for MA lines from Finland, Germany and Israel represented in gold, green and blue, respectively, and are provided, along with estimates of 95% CI, in S17 Table. In (A), gain rates for FB, IA and loss rate for FB are zero.

binomial mixed effects model (Fig 3B; for gains, $\chi^2$ = 5.9, df = 2, $p$ = 0.0514 and losses $\chi^2$ = 12.1, df = 2, $p$ = 0.0024). Post-hoc Tukey HSD tests reveals that Israel genotypes had lower gain rates than Finland genotypes ($p$ = 0.039) and Germany genotypes had greater losses than Israel genotypes ($p$ = 0.0005; S17 Table). In addition to the gain and loss rates, we also calculated a net mutation rate for each genotype (S16 Table) and for each active TE family (S18 Table). These rates range from negative (e.g., $P$ elements in genotype IB are decreasing at a rate of -7.44 x $10^{-4}$ per copy per generation) to positive (e.g., *Penelope* elements increasing at a rate of 9.06 x $10^{-4}$ /copy/ gen in IC), and can even vary for the same TE family among genotypes (e.g., -2.22 and 4.06 x $10^{-4}$ /copy/gen for *Gypsy* elements in GA and FC, respectively).

When selection was not minimized (i.e., in the extant control lineages maintained in large populations in parallel to the MA lines), we only detected 2 gain and 17 loss mutations (Tables 2, S19, and S20). Fitting binomial mixed-effects models, we found that EC lines had significantly lower gain rates ($\chi^2$ = 27.9, df = 1, P < 0.001) and significantly lower loss rates ($\chi^2$ = 10.5, df = 1, P = 0.0012) compared to MA lines, revealing the deleterious effect of TE activity. Furthermore, gain rates in MA lines were 695x higher than in EC lines, compared to loss rates which were only 7.4x higher in MA lines, suggesting that TE gains are much more deleterious than losses (Table 2).

## Validation methods

Rather than perform PCR validation to gauge the sensitivity of our methods, given that each event was of an unknown length, we performed simulations to estimate the false discovery and false omission rate (FDR and FOR) for the four cases of TE events that can occur (Fig 1 and S21 Table). FDRs were relatively low (< 3%) for all four types of mutations (S2 Text and S21 Table), and neither FDRs or FORs varied greatly for TEs of different lengths or for different mutational events (S22 Table). Mutation rates for each type of event in the MA and EC lines adjusted for FDRs can be found in S13 Table. Notably, the fact that the four cases of events are not equally likely (most gains were novel (0 → 1 [n = 62/67]) and most losses were at previously heterozygous sites (1 → 0 [n = 26/28]) is potentially revealing about what proximal mechanisms explain the bulk of TE proliferation and loss (see Discussion). It is important to note, our rate estimates for TE activity likely represent a lower bound. This is, in part, because our analyses focus only on those TEs that could be classified as belonging to one of the five major groups of known TEs (rates for all TEs, classified and unknown, are presented in S23 and S24 Tables).

## TE mutation rates are not correlated with other types of mutation rates

Overall, TE mutation rates in *D. magna* vary intraspecifically among genotypes (Fig 3A) mirroring the high levels of intraspecific variation observed in other mutation rate estimates for this species (see [28, 29]). In terms of frequency per site, TE mutations are intermediate among the other types of mutation examined so far in *D. magna*, (i.e., microsatellite mutation rates are much higher (~$10^{-2}$) and nuclear and mtDNA base substitution rates are much lower (~$10^{-8}$ and ~$10^{-7}$, respectively), on a per site per generation basis). As expected, we observe more events in higher copy number families (S17A Fig). We looked at the relationship between rates of TE gain and loss (and net rates) and the proportion of the genome that is TEs in each genotype and found no correlation (S25 Table), nor do TE rates correlate with base substitution mutation rates (Fig 4A and S25 Table). The only correlation with other mutational processes is between TE mutation rates and gene conversion rates (when plotting only rates for TE events that are likely to be caused by gene conversions [1→0 TE losses and 1→2 TE gains]; ρ = 0.83, $t^7$ = 3.91, P = 0.0058; Fig 4B), although even this predicted correlation is driven largely by one genotype (GB) with high estimates for both rates.

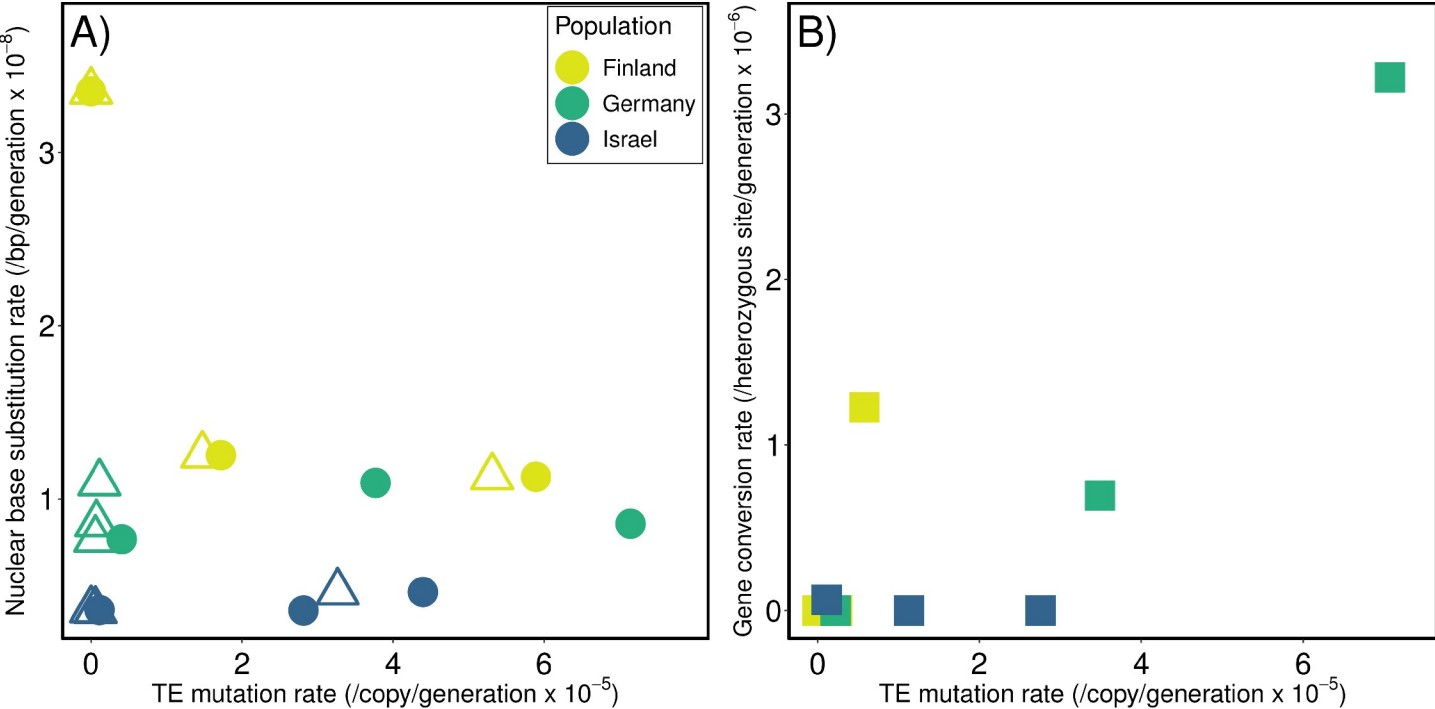

**Fig 4. The relationship between TE, base substitution, and gene conversion mutation rates in *D. magna* MA lines.** (A) Base substitution rates (per bp per generation) plotted against TE mutation rates (per copy per generation). Circles represent the sum of all TE gains and losses, triangles represent only 0→1 TE gains. (B) Gene conversion rates are plotted against TE events that could be caused by gene conversion (the sum of 1→0 TE losses and 1→2 TE gain rates; shown as squares). Points in gold, green, and blue represent rate estimates for genotypes collected from Finland, Germany and Israel, respectively. Base substitution and gene conversion rates are from [29].

## Discussion

Our analyses of TE profiles aim to quantify the levels of intra- and interspecific variation in TE content and mutation rates with and without selection, in order to better understand the mutagenic role of TEs genome-wide over short and long time scales. There are a number of challenges when comparing TE content between lineages or across studies, as differences in repeat content, sequencing technologies, assembly algorithms, software, and pipelines can make standardizing results difficult [36, 37]. In addition, TEs that cannot be classified into any of the major known categories of mobile elements, which are not uncommon, cannot be included in the calculations of family-, class-, or superfamily-specific rates (but see S23 and S24 Tables for rates including 'unknown' TEs; [38]). Furthermore, even if a completely annotated TE library exists, the most commonly used methods for quantifying repeat content in the genome (RepeatMasker [39] versus read-mapping approaches) provide very different estimates of the TE content because the former method relies heavily on assembly quality (see S2 Text). Similarly, our method for measuring TE mutation rates (TEFLoN; [25]) depends on being able to map reads that span gain and loss events, meaning read depth or length can alter the false positive and false negative rates. While we are able to gauge the sensitivity of our methods using simulations, our ability to characterize TEs and detect their movement is likely to continue to improve with technological and bioinformatic advances (S21 Table).

Previous work on *Daphnia* TEs (e.g., [27, 40, 41]) utilized their unique reproductive mode (typically, cyclical parthenogenesis [asexual reproduction with occasional bouts of sex], but with the repeated evolution of obligate asexuality) to explore an early and frequently posed question about how TEs proliferate via sex [42]. These studies and those in other species that

can reproduce with and without sex have painted a complex picture: some TEs exhibit different patterns of proliferation among sexuals and asexuals (e.g., in *D. pulex* [43]), but this is not always the case (e.g., in yeast [[44] reanalysis of data from [45]]). Even though most *Daphnia* can reproduce sexually, they can be propagated in the lab exclusively via asexually-produced clonal offspring, allowing us to estimate rates of TE gain and loss without the complicating influence of sex, unlike TE studies in *Drosophila* (reviewed in [46]). Although the lineages in this study were reared without sex during the MA experiment, the 9 starting genotypes of *D. magna* originally collected from Finland, Germany, and Israel (Fig 2A) have, historically, experienced quite varied environmental regimes (S1 Table), likely impacting the frequency of sexual reproduction in the past and/or influencing effective population sizes. The differences in mean temperatures, temperature ranges, light exposure, and drought conditions across the latitudinal gradient surveyed here helps provide a glimpse of the intraspecific variation in mutation rates typically overlooked by most studies estimating mutation rates for only one or a few genotypes. It is known, for example, that Finnish genotypes experience freezing temperatures and yearly dry downs, whereas German genotypes experience only freezing temperatures and genotypes from Israel experience only seasonal dry downs [47]. These ecological differences, paired with different rates of recombination [48, 49], could result in a historical selection regime tolerant of different mutation rates if, for example, frequent population bottlenecks in Finnish rock pools maximize drift relative to selection. Ultimately, our quantification of accumulated TE content (over long time periods) and rates of TE movement (over short time periods) in the *Daphnia* genome will help disentangle the mutational input provided by TEs from the evolutionary forces that subsequently shape the repetitive portion of the genome.

## Long-term patterns of TE accumulation do not correspond to short-term mutation rates

Overall, TE content, in terms of abundance, is very similar across genotypes from the three populations sampled for this study (Fig 2B). Elements from the *Gypsy* superfamily of LTRs (Class 1) are the most numerous, as has been reported in the congener, *D. pulex* (Rho et al. 2010), which has more TEs overall than *D. magna* (Table 1) even though *D. magna* has a larger genome (as measured by flow cytometry, *D. magna* = 0.30 pg and *D. pulex* = 0.23 pg [33]). Despite these similarities in patterns of TE abundance, patterns of insertion site polymorphism (differences among individuals in terms of which specific sites are occupied by TEs of a given family) make all three populations readily distinguishable (Fig 2C and S8 Table), which begs the question—how much do mutation rates for TEs differ intraspecifically in *Daphnia*?

Based on over 100 observed events in our multi-year MA experiments, we were able to estimate rates of gain and loss for each type of TE mutation (Table 2). Rates of gain and loss in *D. magna* are similar (1.4 and 1.7 x $10^{-5}$ per copy per generation, respectively; Table 2), but they vary widely among genotypes and populations (Fig 3A and 3B) and among TE families (Table 3). The majority of the gains observed are novel gains ($0 \rightarrow 1$ gains; Fig 1), most likely resulting from insertions of TEs either excised from elsewhere in the genome (in the case of cut-and-paste elements) or retrotransposed (in the case of Class I elements, such as *Gypsy*), rather than $1 \rightarrow 2$ gains which can result from homolog-dependent DNA repair [50]. The majority of loss events were at positions that were initially heterozygous ($1 \rightarrow 0$), again a pattern expected based on mechanism since both DNA repair and gene conversion events could "reconstitute" a TE lost due to excision or deletion at an ancestrally homozygous site. A genome-wide assay of TE mutation rates in *Drosophila* showed insertions far outnumber deletions, but in flies the per copy per generation rates differ significantly, with insertions higher (~$10^{-9}$) than deletion rates (~$10^{-10}$), and much lower rates overall compared to those observed here [25].

Little is known about intraspecific variation in TE mutation rates in other animal species, even though there have been several large-scale studies of their polymorphism (e.g., [51–53]). Among *D. magna* genotypes, rates ranged from a high gain bias in one genotype from Finland (FA; $5.3 \times 10^{-5}$ per copy per generation) to a deletion bias in one genotype from Germany (GB; $-5.5 \times 10^{-5}$ per copy per generation; S16 Table), with the highest number of events overall occurring in a single genotype (FC; S16 Table) in a single family (*Gypsy*; n = 51; S18 Table). Looking across families of TEs, populations are distinct in their rates, with Finland exhibiting higher rates of gain overall, Germany exhibiting high rates of loss overall, and Israel exhibiting gains and losses with almost equal frequency resulting in the lowest net rates overall (Fig 3B and S16 and S17 Tables). Thus, while genotype-specific rates of mutation surely introduce variable levels of TE-related genetic variation in these lineages, evolutionary forces acting at the population-level likely explain the consistent overall abundance of TEs (Fig 2B) and distinctive patterns of insertion site polymorphism (Fig 2C).

Ultimately, the lack of correspondence between the variable mutation rates and the consistent patterns of TE accumulation across the 9 genotypes suggests natural selection may prevent TEs from over-running the genome long-term. Evidence in support of selection against TE activity from this study was our observation of much lower rates in control lines (where lineages were maintained in large population sizes) compared to MA lines (where selection is minimized by propagating lines via single-progeny descent), suggesting that TE mutations, especially gains, are highly deleterious (Table 2). Early papers on rates of TE activity posited that high copy number families might even evolve lower transposition rates because of the deleterious effects of TE insertions (much like parasites evolve to be less virulent; [54]), however the relationship we observe between per copy per generation rates of mutation and abundance in the genome observed is weak (S17A Fig), with no clear downward trend even for high copy number families (S17B Fig) or with rates of gain (S17C Fig).

Looking specifically at the most abundant family with the most mutation events, *Gypsy*, we see rates of gain and loss can vary greatly among genotypes (Fig 5A) and, in this case, the variation is reflected in the long-term patterns of insertion site polymorphism (Fig 5B) and abundance (Fig 5C). While the patterns reflect a mixture of active and inactive elements, some

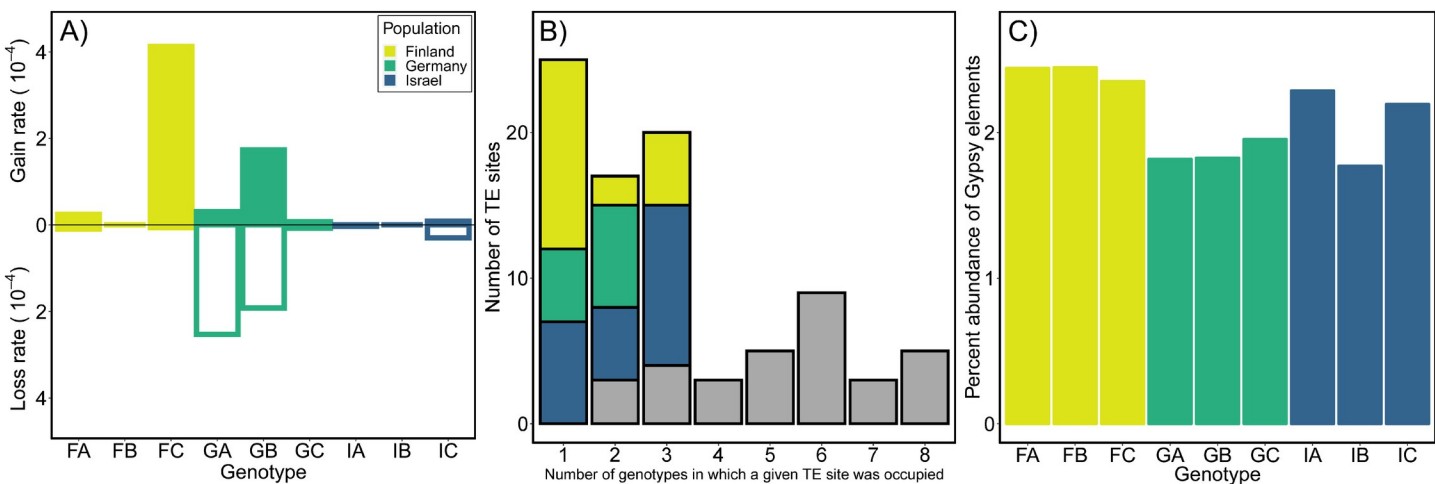

**Fig 5. Rates of insertion site polymorphism and abundance for *Gypsy* family TEs in 9 genotypes of *D. magna* from Finland (gold), Germany (green), or Israel (blue).** (A) Mean gain and loss rates (per copy per generation) for each genotype. Gain rates for FB, IA, IB and loss rates for FB, IB are zero. (B) Colored bars indicate the number of genotype-specific polymorphic sites (singletons; x = 1) or population-specific sites (when x = 2 and x = 3), grey bars represent sites where elements are shared across populations (reference genome used for this analysis was FASC). (C) Percent abundance in the genome for each genotype estimated using RepeatMasker.

population-specific trends, which have been reported previously for *Gypsy* elements [55], are notable. Specifically, high rates of gain in Finnish genotypes could explain a non-significant trend in terms of singletons (excess in Finland [n = 13] compared to Germany [n = 5] or Israel [n = 7]; G = 4.01, df = 2, P = 0.13) or the higher percent abundance of *Gypsy* elements in Finnish clones compared to Germany (Fig 5C). Future studies with additional genotypes and populations and a longer mutation accumulation experiment will be necessary to determine if the patterns of TE accumulation reflect the mutational variation, as suggested by the data for this large TE family, or if evolutionary forces mute the variation introduced by TE movement, as observed when looking across all families of elements.

## Rates of TE gain and loss do not correlate with other mutation rates

Base substitution mutation rates (bsMRs) are the most frequently estimated, and are used broadly in models and discussion of the mutation rate in evolutionary biology. Although they are the most commonly studied, bsMRs are not necessarily representative of mutation rates for other categories of mutation, nor are they likely to generate they greatest amount of genetic variation [56]. Microsatellites are known to be highly mutable (reviewed in [57]) and the average genome-wide rates of mutation at these loci in these genotypes of *D. magna* are several orders of magnitude higher ($\sim 10^{-2}$; [28]) than the TE mutation rates we report here ($\sim 10^{-5}$). The bsMRs we reported for *D. magna* were the highest and most variable direct estimates reported in animals so far using an MA approach ($\sim 10^{-7}$ and $\sim 10^{-9}$ for the mtDNA and nucleus, respectively; [29]), but are also several orders of magnitude lower than the overall TE mutation rates we report. Evolutionary theory aimed at explaining how mutation rates evolve does not specify mutation types, however, thus we would expect that lineages with relatively high rates of mutation in one category would have high mutation rates for other types of mutation as well. The data do not support this prediction, as there is no correlation between TE mutation rates and bsMRs across the 9 genotypes (Fig 4A and S25 Table). Rates of gene conversion, however, do positively correlate with TE rates when based on those events that can be produced by gene conversion as predicted (Figs 1 and 4B and S25 Table).

While there is no positive linear correlation among mutation rates for different types of mutations (comparing TEs and base substitutions) across all 9 genotypes (Fig 4A), it is interesting to note that, in our MA experiment, genotypes from Finland have the highest rates of TE gain (and gains are more deleterious than losses), the highest rates of microsatellite deletions [28], the highest rates of base substitution among the three populations assayed [29], and the highest rates of mutations causing structural variation (e.g., insertions and deletions) [30]. These commonalities among our direct estimates based on rearing animals in a common laboratory environment point to the historical selection regime due to population genetic constraints or the frequency of recombination, rather than mutagens in the atmosphere, as an explanation for higher rates of deleterious mutation in the Finnish genotypes. Alternatively, this pattern could result if selection on DNA repair mechanisms, as opposed to the mechanisms causing mutations, is more influential. In contrast, genotypes from Israel consistently exhibit the lowest net rates of TE mutation, microsatellite mutation, and base substitutions of the three populations assayed.

## Conclusions

Few direct estimates of TE mutation rates have been published outside of classic model organisms in genetics and our own species (e.g., from *Drosophila* [58], *Arabidopsis* [59], and human [60]), however adding to this list and quantifying levels of intraspecific rate variation is key for understanding how rates evolve. Furthermore, investigating the correspondence between TE

mutation rates and long-term patterns of accumulation is essential for understanding genome evolution and finding solutions to long-standing puzzles, such as the C-value paradox [61, 62]. Finally, differences in rates among categories of mutations or genomic compartments (e.g., [63, 64]) pose a challenge to evolutionary theory, and require that we expand our investigation of mutation rates beyond base substitution rates in the nuclear genome [65]. Our study shows rates of TE mutation are high, variable, and uncorrelated with rates for other categories of mutation, making them important engines of change generating genetic variation worthy of further investigation. Future work aimed at understanding the causes and consequences of mutation rate variation within populations and species, the heritability and evolvability of mutation rates for different types of mutation, and the significance of the mobilome for generating genetic variation are necessary to improve our understanding of how mutation rates evolve over time and space.

## Methods

### Study system

The *D. magna* genotypes used in this experiment were provided by Dieter Ebert and are part of a collection of samples from across the species range. Genotypes were selected from populations along a latitudinal gradient (Finland, Germany, and Israel) in order to sample individuals originating from a broad range of environments. Different maximum and mean temperatures and photoperiods (S1 Table), both of which can also result in fluctuating habitat sizes [66], are represented along the gradient.

### Experimental design

Three genotypes from each of three populations (Finland, Germany, and Israel) were used to initiate laboratory stocks. From these lab stocks, starting controls (SCs) were selected (immediate descendants of which were frozen and sequenced) for each of the 9 genotypes. From the SCs, mutation accumulation (MA) lines (n = 5–12 per genotype; total of 66) and large population controls (extant controls [ECs]; n = 2 per genotype; total of 18) were initiated and propagated in parallel. Tissue from each line (MAs and ECs) was frozen after the mutation accumulation period; the average number of generations across MA lines was 12 and the experiment ran for approximately 30 months in total (S26 Table; see S1 Text for additional details).

The MA and EC lines from each genotype were maintained as single individuals or large populations in 250 mL beakers containing 175–200 mL or 3.5 L jars containing 3 L of Aachener Daphnien Medium (ADaM [67]), respectively. All lines were maintained under a constant photoperiod (16L:8D) and temperature (18˚C), and fed the unicellular green alga *Scenedesmus obliquus* (2–3 times per week *ad libitum*). While selection is permitted to act in the large population ECs, the single-progeny descent used to propagate the MA lines maximizes chance and minimizes selection, and thus allows for the accumulation of mutations. The experimental protocols used here have been described previously [28, 29].

### DNA extraction and sequencing

At the end of the mutation accumulation period, the 9 SCs, 66 MA lines, and 18 ECs were sequenced (Illumina) to assess the TE content in the original genotypes (SCs), to quantify TE mutation rates (MA lines), and to compare to laboratory-reared lines where selection is not minimized (ECs). Five asexually-produced clonal individuals from each SC line, all derived MA lines, and the extant control lines were flash frozen for DNA extractions (see S1 Text for

details). Libraries were used to generate approximately 50x depth of coverage genome-wide for each sample. Paired-reads from SC lines were then used to construct reference-guided assembles for each of the 9 genotypes (see S2 and S17 Tables for genome assembly statistics and S1 Text for assembly methods).

## Characterizing TE content

A custom *D. magna* TE consensus library was created from a concatenated file of the 9 reference-guided assemblies from the SC for each genotype using RepeatModeler v1.0.11 [68] and used to mask each assembly using the slow search setting of RepeatMasker v4.1.0 [39]. We clustered elements in the TE library that exhibited $\geq$ 98% nucleotide identity over their full length to a longer sequence in the library using cd-hit-est v4.8.1 [69], yielding a non-redundant TE library containing full and partial TE copies (S1 Data). The non-redundant TE library was then used to determine the abundance, length, percent occupancy, insertion site polymorphism, and pairwise divergence for all categorized TEs in each assembly (see S1 Text for details), and in some cases analyses were performed using both categorized and 'unknown' TEs. To compare TE abundance and diversity to the congener *D. pulex*, we utilized the publicly available reference assembly PA42 (https://www.ncbi.nlm.nih.gov/bioproject/307976). The quality of our *D. magna* assemblies were similar to that of the *D. pulex* assembly (S2 Table).

## TE mutation rate estimation in MA lines

We used TEFLoN v0.4 [25] to identify active TEs in the MA lines (see S1 Text for details). There are two types of TE gain mutations (0$\rightarrow$1 and 1$\rightarrow$2) and two types of TE loss mutations (2$\rightarrow$1 and 1$\rightarrow$0) that can be observed based on whether the ancestor (SC) was homozygous, heterozygous or lacked a TE (an "absence allele") at a given site relative to the status in the descendant MA line (e.g., if the SC was heterozygous and experienced a gain, it would be classified as a 1$\rightarrow$2 gain event in the MA line; Fig 1). Our ability to detect these different events is not uniform, however, thus we used a series of filtering steps and simulations to assess the support for each observed event and to assess the sensitivity of our methods (see S1 Text). Family-specific mutation rates for each of the four mutation types were calculated using $N_m / (N_{SC}*G)$, where $N_m$ represents that number of sites that experienced a particular mutation event, $N_{SC}$ represents the initial copy number of that TE family in the SC line, and G represents the number of MA generations. For a full description of our estimates of our false discovery and false omission rates and our simulations, see the S1 Text.

## Statistical analyses

Statistical analyses were performed in R [70]. Family-specific TE mutation rates for a particular genotype was estimated by averaging across MA lines. Rates of a particular mutation type (0$\rightarrow$1 gain, 1$\rightarrow$2 gain, 1$\rightarrow$0 loss, 2$\rightarrow$1 loss) of an MA line were estimated by averaging that rate across all TE families. Rates of a particular mutation type for a genotype were estimated by averaging that rate across MA lines. Confidence intervals for mutation rates were estimated by bootstrapping across MA lines 10000 times. Details on all statistical test are included in S1 Text and all code for data processing and analysis is available at https://github.com/EddieKHHo/DaphiaMagna_MA_TE.

## Supporting information

**S1 Text. Supplementary methods.**
(DOCX)

**S2 Text. Supplementary results.**
(DOCX)

**S1 Data. TE library constructed by RepeatModeler using the 9 refernece assemblies of *Daphnia magna*.**
(FASTA)

**S1 Table. Collection data for the 9 starting genotypes of *Daphnia magna* (FASC, FBSC, FCSC, GASC, GBSC, GCSC, IASC, IBSC, ICSC) used in this study.**
(XLSX)

**S2 Table. Assembly statistics for the 9 starting genotypes of *Daphnia magna* collected originally from Finland (FASC, FBSC, and FCSC), Germany (GASC, GBSC, GCSC), and Israel (IASC, IBSC, and ICSC) and one genotype of *Daphnia pulex* (PA42 version 4.1) for which sequence data were publicly available (https://www.ncbi.nlm.nih.gov/bioproject/307976).**
(XLSX)

**S3 Table. ANOVA results for the log TE percent abundance between nine genotypes of *D. magna*.**
(XLSX)

**S4 Table. TE content for each starting genotype of *Daphnia magna* collected originally from Finland (FASC, FBSC, and FCSC), Germany (GASC, GBSC, GCSC), and Israel (IASC, IBSC, and ICSC) compared to *D. pulex*, including amount (megabases [Mb]), percent of assembly, and number of elements according to RepeatMasker results.**
(XLSX)

**S5 Table. Mean proportional abundance of each TE family or superfamily in *Daphnia magna* (averaged across 9 starting genotypes originally collected from Finland, Germany, and Israel) using two different methods (read mapping to a repeat library and repeat masking with a repeat library).**
(XLSX)

**S6 Table. ANOVA results for the log TE percent abundance between *D. magna* and *D. pulex*.**
(XLSX)

**S7 Table. Number of sites and percent polymorphism (means, in bold) for each TE superfamily in each of the genomes sequenced from the 9 starting genotypes of *Daphnia magna* collected originally from Finland (FASC, FBSC, and FCSC), Germany (GASC, GBSC, GCSC), and Israel (IASC, IBSC, and ICSC).** Analyses were also performed using each possible reference genome using all sites that passed filters for each (regular font).
(XLSX)

**S8 Table. K-means clustering of principal component axes from a Principal Component Analysis of TIPs identified in analyses using each of the nine genotypes of *Daphnia magna* as reference assemblies.**
(XLSX)

**S9 Table. Proportion of singleton and population-specific sites among three genotypes of *Daphnia magna* each from Finland, Germany, and Israel for all TE families combined.**
(XLSX)

**S10 Table. Polymorphism levels and proportion of population-specific sites among three genotypes of *Daphnia magna* each from Finland, Germany, and Israel for the seven most abundant TE superfamilies.**
(XLSX)

**S11 Table. Abundance and mean pairwise divergence of active and inactive TE superfamilies in the 9 starting genotypes of *Daphnia magna* collected originally from Finland (FASC, FBSC, and FCSC), Germany (GASC, GBSC, GCSC), and Israel (IASC, IBSC, and ICSC), where active families are those found to exhibit new mutations in the mutation accumulation experiment conducted as part of this study.**
(XLSX)

**S12 Table. Mean pairwise divergence of TE superfamilies shared by *Daphnia magna* (averaged across nine genotypes) and *D. pulex* (PA42 [PRJNA307976]).**
(XLSX)

**S13 Table. Mean and adjusted rates for each type of mutation for MA annd EC lines of each *D. magna* genotype.**
(XLSX)

**S14 Table. List of all TE mutation events in MA and EC lines of *D. magna*.**
(XLSX)

**S15 Table. Mutation count and rate for each MA and EC line descending from the 9 starting genotypes of *Daphnia magna* collected originally from Finland (FA, FB, and FC), Germany (GA, GB, GC), and Israel (IA, IB, and IC).**
(XLSX)

**S16 Table. Post-hoc Tukey HSD tests of binomial mixed effect models on the effect of population on gain and loss rates for *D. magna* MA lines from Finland, Germany and Israel.**
(XLSX)

**S17 Table. Estimates of mean gain, loss, total and net rates averaged across TE families and for only *Gypsy* elements based on MA lines derived from 9 starting genotypes of *Daphnia magna* collected originally from Finland (FA, FB, and FC), Germany (GA, GB, GC), and Israel (IA, IB, and IC).**
(XLSX)

**S18 Table. Estimates of mean gain, loss and net rates (per copy per generation) for each TE superfamily averaged across MA lines derived from 9 starting genotypes of *Daphnia magna* collected originally from Finland (FA, FB, and FC), Germany (GA, GB, GC), and Israel (IA, IB, and IC).**
(XLSX)

**S19 Table. Estimates of mean gain and loss rates (per copy per generation) averaged across TE families in extant control lines derived from 9 starting genotypes of *Daphnia magna* collected originally from Finland (FA, FB, and FC), Germany (GA, GB, GC), and Israel (IA, IB, and IC).**
(XLSX)

**S20 Table. Number of events and mean rates of gain and loss (per copy per generation) for each TE superfamily in extant control lines derived from 9 starting genotypes of *Daphnia magna* collected originally from Finland (FA, FB, and FC), Germany (GA, GB, GC), and**

Israel (IA, IB, and IC).
(XLSX)

**S21 Table. False discovery and false omission rates for each type of TE mutation across all simulations.**
(XLSX)

**S22 Table. False discovery and false omission rates for each type of TE mutation for simulations with different TE minimum lengths.**
(XLSX)

**S23 Table. Count of gains and losses when using TE libraries with and without unknown repeats to estimate mutation rates in MA lines derived from 9 starting genotypes of *Daphnia magna* collected originally from Finland (FA, FB, and FC), Germany (GA, GB, GC), and Israel (IA, IB, and IC).**
(XLSX)

**S24 Table. Mutation rate estimates for the analyses performed with and without unknown repeats in the repeat library using whole genome sequence data from MA lines derived from 9 starting genotypes of *Daphnia magna* collected originally from Finland (FA, FB, and FC), Germany (GA, GB, GC), and Israel (IA, IB, and IC).**
(XLSX)

**S25 Table. Correlations between proportions of the genome comprised of TEs and TE mutation rates.** Proportions (top) of different TE types in the genome estimated using the read-mapping approach as they correlated with rates of gain, loss, and net rates for TE mutations. Mutation rates (bottom) for other mutation types as they correlate with TE mutation rates (different subsets shown). Base substitution rates (per nucleotide per generation), gene conversion rates (per heterozygous site per generation) and microsatellite mutation rates (absolute value of the mutation rate per copy per generation and net copy number change per copy per generation) are from Ho et al. (2019, 2020).
(XLSX)

**S26 Table. Number of generations and statistics for paired-end sequencing reads generated from each starting control (SC), mutation accumulation (MA, and extant control (EC) line sequenced from each the 9 starting genotypes of *Daphnia magna* collected originally from Finland (FASC, FBSC, and FCSC), Germany (GASC, GBSC, GCSC), and Israel (IASC, IBSC, and ICSC).**
(XLSX)

**S27 Table. Analysis of genetic relatedness among three populations of *Daphnia magna* based on pairwise genetic distances from single nucleotide variants.**
(XLSX)

**S28 Table. Pearson correlations of TE insertion rates against median depth of coverage for MA lines of each genotype of *Daphnia magna*.**
(XLSX)

**S29 Table. Presence and absence of TE for each starting genotype of *Daphnia magna* at polymorphic TE sites.**
(XLSX)

**S1 Fig. Principal Component Analysis based on the presence/absence of TEs when using each of the nine reference assemblies.** Variance explained by principal components 1 and 2

are displayed on the axes. The reference assembly used is indicated on the top of each plot. Genotypes from Finland, Germany and Israel are colored in gold, green, blue, respectively.
(TIF)

**S2 Fig. Number of polymorphic TE sites occupied across the 9 genotypes.** The left bar represents the number of singletons (sites occupied in only one genotype) for each population (gold, green and blue for Finland, Germany and Israel, respectively). Colored portions of bars in x = 2 and x = 3 represent sites occupied in 2 and 3 genotypes, respectively, when from the same population. Grey portions of each bar represent the number of sites that were occupied in ≥2 genotypes that were not population-specific. The reference assembly used is indicated on the top of each plot.
(TIF)

**S3 Fig. Proportion of singletons TEs in each population for analyses using different reference genomes.** Gold, green and blue represents singletons specific to genotypes in Finland, Germany and Israel, respectively. The proportion of singletons belonging to each population was not significantly different when using different reference genomes ($\chi^2$ = 8.8, df = 16, P = 0.92).
(TIF)

**S4 Fig. Pairwise divergence of TEs in the *D. magna* FASC assembly for TE families that are active within *D. magna* MA lines.**
(TIF)

**S5 Fig. Pairwise divergence of TEs in the *D. pulex* reference genome (PA42) for families that are active within *D. magna* MA lines.**
(TIF)

**S6 Fig. Pairwise divergence of DNA/*Academ-1* for all nine reference genomes of *D. magna* originally collected from Finland (FASC, FBSC, and FCSC), Germany (GASC, GBSC, and GCSC), and Israel (IASC, IBSC, and ICSC).**
(TIF)

**S7 Fig. Pairwise divergence of DNA/*CMC-EnSpm* for all nine reference genomes of *D. magna* originally collected from Finland (FASC, FBSC, and FCSC), Germany (GASC, GBSC, and GCSC), and Israel (IASC, IBSC, and ICSC).**
(TIF)

**S8 Fig. Pairwise divergence of DNA/*hAT-Ac* for all nine reference genomes of *D. magna* originally collected from Finland (FASC, FBSC, and FCSC), Germany (GASC, GBSC, and GCSC), and Israel (IASC, IBSC, and ICSC).**
(TIF)

**S9 Fig. Pairwise divergence of DNA/*P* for all nine reference genomes of *D. magna* originally collected from Finland (FASC, FBSC, and FCSC), Germany (GASC, GBSC, and GCSC), and Israel (IASC, IBSC, and ICSC).**
(TIF)

**S10 Fig. Pairwise divergence of DNA/*PIF-ISL2EU* for all nine reference genomes of *D. magna* originally collected from Finland (FASC, FBSC, and FCSC), Germany (GASC, GBSC, and GCSC), and Israel (IASC, IBSC, and ICSC).**
(TIF)

**S11 Fig. Pairwise divergence of LINE/*I* for all nine reference genomes of *D. magna* originally collected from Finland (FASC, FBSC, and FCSC), Germany (GASC, GBSC, and GCSC), and Israel (IASC, IBSC, and ICSC).**
(TIF)

**S12 Fig. Pairwise divergence of LINE/*Penelope* for all nine reference genomes of *D. magna* originally collected from Finland (FASC, FBSC, and FCSC), Germany (GASC, GBSC, and GCSC), and Israel (IASC, IBSC, and ICSC).**
(TIF)

**S13 Fig. Pairwise divergence of LTR/*DIRS* for all nine reference genomes of *D. magna* originally collected from Finland (FASC, FBSC, and FCSC), Germany (GASC, GBSC, and GCSC), and Israel (IASC, IBSC, and ICSC).**
(TIF)

**S14 Fig. Pairwise divergence of LTR/*Gypsy* for all nine reference genomes of *D. magna* originally collected from Finland (FASC, FBSC, and FCSC), Germany (GASC, GBSC, and GCSC), and Israel (IASC, IBSC, and ICSC).**
(TIF)

**S15 Fig. Pairwise divergence of LTR/*Pao* for all nine reference genomes of *D. magna* originally collected from Finland (FASC, FBSC, and FCSC), Germany (GASC, GBSC, and GCSC), and Israel (IASC, IBSC, and ICSC).**
(TIF)

**S16 Fig. Mean divergence of TE copies plotted against mean base substitution rates for each starting genotype.** Divergence averaged across all TE families, only active TE families, and only inactive TE families are plotted as filles squares, filled circles, and empty circles, respectively.
(TIF)

**S17 Fig. Relationship between TE content and mutation rates.** Percent abundance (log scale) of each TE family averaged across genotypes in *D. magna* plotted against (A) number of mutation events for each TE family and (B) gain and loss rates for each active TE family. Gain rates for DNA/CMC-EnSpm, DNA/P, LTR/DIRS, LINE/I and loss rates for DNA/Academ-1, LINE/Penelope are not shown because there were zero mutation events. (C) Percent of the genome occupied by TEs for each assembly plotted against the TE gain (black), loss (white) and net (grey) rates averaged across all families and MA lines. Percent abundance of TEs was estimated using the read mapping approach.
(TIF)

## Acknowledgments

We would like to thank Maia J. Benner, Dana Howe, Dee Denver, Dieter Ebert, Peter Fields, and Jeremy Coate for supplying animals, technical assistance, resources/support, and helpful feedback. The map in Fig 2A was made with BioRender.com.

## Author Contributions

**Conceptualization:** Sarah Schaack.

**Data curation:** Jaclyn Calkins, Leigh C. Latta IV.

**Formal analysis:** Eddie K. H. Ho, Emily S. Bellis, Jeffrey R. Adrion, Sarah Schaack.

**Funding acquisition:** Sarah Schaack.

**Investigation:** Eddie K. H. Ho, Jaclyn Calkins, Sarah Schaack.

**Methodology:** Eddie K. H. Ho, Emily S. Bellis, Sarah Schaack.

**Project administration:** Leigh C. Latta IV, Sarah Schaack.

**Resources:** Eddie K. H. Ho, Jeffrey R. Adrion.

**Software:** Jeffrey R. Adrion.

**Supervision:** Sarah Schaack.

**Validation:** Eddie K. H. Ho.

**Visualization:** Eddie K. H. Ho.

**Writing – original draft:** Eddie K. H. Ho, Sarah Schaack.

**Writing – review & editing:** Eddie K. H. Ho, Emily S. Bellis, Jaclyn Calkins, Jeffrey R. Adrion, Leigh C. Latta IV, Sarah Schaack.

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
