## [Decision Letter · Decision Letter 0]

7 Sep 2021

Dear Dr Schaack,

Thank you very much for submitting your Research Article entitled 'Engines of change: Transposable element mutation rates are high and variable within Daphnia magna' to PLOS Genetics.

The manuscript was fully evaluated at the editorial level and by three independent peer reviewers. The editors and all three reviewers were enthusiastic about your contribution, but the reviewers have identified some concerns that we ask you address in a revised manuscript.

We therefore ask you to modify the manuscript according to the review recommendations.  Of these, clarification regarding some statistical issues (bootstrap versus mixed model), and further details regarding genome assembly are essential.  I would also re-iterate that headings within the results section would aid the reader and really help people grasp the "take-home" messages of your study.  While we highlight these three points, all three reviewers had excellent (although relatively minor) comments and suggestions; please address each of their comments in the revision.  

[LINK]

Yours sincerely,

Lindi Wahl

Associate Editor

PLOS Genetics

Kirsten Bomblies

Section Editor: Evolution

PLOS Genetics

Reviewer's Responses to Questions

**Comments to the Authors:**

Reviewer #1: In this manuscript, Ho, Bellis, et al. investigate transposable element variation and mutation rates in different genotypes of Daphnia magna. The authors explain the different ways TE copy number may change in Daphnia, which is useful to the reader because of the unique aspect of asexuality. The authors find that mutation rates are variable between lines, and interestingly several lines show a directional bias. They also propagated populations of the same genotypes where selection occurred and show that transposable element mobilization is constrained. Overall, the manuscript’s data support the authors’ conclusions which represent a strong contribution to the field. However, the manuscript could be improved in clarity and organization. There are also a few details missing from the manuscript, and discrepancies, which need to be clarified for full confidence in the paper’s findings and conclusions.

1. The results section was challenging to read as many results were listed and it was difficult to extract take-home messages. I suggest further subdividing the results section, and making subsection titles more informative and summarizing the results. For example, the authors may consider subheadings similar to these:

Line 167:

Characterizing TE content in Daphnia

Line 187:

Variation in TE activity on a long-term scale

Line 214:

Estimated rates of TE loss and gain using mutation accumulation lines

Line 236:

TE activity is under selective constraint

Line 245:

Validation of methods for detecting TE insertion mutations

Line 259:

TE mutation rates are not correlated with other types of mutation

2. The authors use simulations to estimate false positive and false negative rates. They mention in the discussion that read depth can alter the false positive and negative rates, however it seems this was not factored into the simulations. Please clarify if there was a correlation between depth and number of insertions detected. Why was 50x depth used for the simulations and not the empirical depth, which will vary between different lines? I could not find any mention of the empirical depth in the manuscript or supplement.

3. Confidence intervals are overlapping for TE losses between ECs and MA lines in Table 2. Why are the bootstrap results presented when a mixed effect model was referred to in the main text? Why is the mixed effect model more suitable than the bootstrap, as the two methods result in a discrepancy in significance for TE loss rates?

4. Please include a clarification of a couple details about the EC lines. How was the number of generations estimated and what is the confidence of this estimate (e.g. there may be overlapping generations)? Please briefly explain the limitations of your sampling approach of the EC lines, as you may not be capturing variation within the population.

5. L251 – please clarify that you are referring to the empirical data, and FDR/FOR didn’t greatly vary between the mutation types

6. Why were the EC rates not adjusted in the table S9? Please include explanation in the simulations section and/or the table legend.

7. L281-283: Each unknown element identified by RepeatModeler may represent a family, and in theory could have a family-specific mutation rate. I think you should say class or superfamily -level specific rates.

8. Please try to reword the sentence on lines 401-403 to increase clarity.

9. In comparison of D. pulex and D. magna TE content, why is the RepeatMasker method used when you state it is a poor method for comparing different assemblies especially those of different qualities?

Reviewer #2: This is a very well-written article that sheds light not only on the abundance and diversity of transposable elements (TEs) in populations of Daphnia magna across a latitudinal gradient and how this compares with that of the closely related species D. pulex, but also on estimating mutation rates with and without selection across populations. Determining whether transposition rates vary across populations is a fundamental and still open question in the field, and the results presented in this manuscript do contribute to answer it.

I have a few suggestions for the authors to consider.

One of the metrics used to estimate overall abundance is affected by the quality of the assemblies, as mentioned by the authors. The authors apparently used previous available assemblies. Would the authors consider adding a few lines about the quality of the assemblies? Is it comparable across genomes? Are these assemblies based on long-reads? how the variation in assembly impacts the abundance estimates? Part of this analysis is currently in supplemental material, I think it deserves a brief mention in the results section as well. Along the same lines, how does the assemblies of D. magna and D. pulex compare?

Line 175. Maybe mention what does “t8” stand for?

Line 205 Do you mean “higher MPD in TE families that were currently active”?

Line 482. Was redundancy not remove from the TE library? How can this affect the annotations?

Line 529. I would encourage the authors to upload their TE library to a public repository such as Dfam so that it is more easily accessible to potential users rather than providing (or additional to providing it) as a supplemental data file.

Figure 1. Why losses from 1 to 0 cannot be due to cut and paste transposition?

Figure 2. Increase contrast of the map so that sampling collections are more conspicuous.

Figure 3. Are the gain and loss rates of FB zero? Maybe mention it in the legend if that is the case. Same for figure 5A, FB an IB.

Typos in lines 115, 550

Reviewer #3: This paper quantifies the transposable element activity in a 30-month mutation-accumulation experiment involving 9 genotypes of Daphnia magna. The results are compared to large populations of the same genotypes, in which selection is allowed to act. A total of 95 mutation events are recorded in the MA lines, 70 of which involved gypsy-elements. The mutation rates vary widely between lines and are much lower in the large control populations. I find this an interesting and well-presented study.

I have only minor comments:

line 65-68. Inducing structural variation in the genome might be added to this list.

line 196-199. I am not sure I see this distinction (you use “or”). Both types of processes operate at the same time, I would think.

line 205. I think this should be “higher”. Lower is what you expect.

line 302. Chen and Zhang reanalysed Bast et al. 2019, not 2016.

line 404-405. This is an important point, worth emphasising. It extends to TE family differences.

line 418-422. Perhaps selection is acting on DNA repair mechanisms?

line 427-428. I find it difficult to get a sense of how the Daphnia TE mutation rates compare to these other organisms, except Drosophila, which is mentioned on line 347).

line 477. Please mention briefly the sequencing technique (Illumina short reads) and assembly method (reference-guided) here.

line 512. Sentence truncated.

**Have all data underlying the figures and results presented in the manuscript been provided?**

Reviewer #1: **No: **Code is accessible on the Github page, however I was unable to access data at the accession PRJNA658680 at NCBI. Please ensure the sequencing data is publicly available before full acceptance.

Reviewer #2: **No: **Some will be provided upon acceptance

Reviewer #3: Yes

PLOS authors have the option to publish the peer review history of their article (what does this mean?). If published, this will include your full peer review and any attached files.

Reviewer #1: No

Reviewer #2: No

Reviewer #3: No

---

## [Editor Report · Decision Letter 1]

16 Sep 2021

Dear Dr Schaack,

We are pleased to inform you that your manuscript entitled "Engines of change: Transposable element mutation rates are high and variable within Daphnia magna" has been editorially accepted for publication in PLOS Genetics. Congratulations!  Thanks for your care in putting the revisions together, which all seem clear and reasonable.  This is a strong contribution and we're glad that the review process made it even stronger.

Yours sincerely,

Lindi Wahl

Associate Editor

PLOS Genetics

Kirsten Bomblies

Section Editor: Evolution

PLOS Genetics

**Data Deposition**

http://datadryad.org/submit?journalID=pgenetics&manu=PGENETICS-D-21-01012R1

**Press Queries**

---

## [Editor Report · Acceptance letter]

21 Oct 2021

PGENETICS-D-21-01012R1 

Engines of change: Transposable element mutation rates are high and variable within </i>Daphnia magna</i> 

Dear Dr Schaack, 

We are pleased to inform you that your manuscript entitled "Engines of change: Transposable element mutation rates are high and variable within </i>Daphnia magna</i>" has been formally accepted for publication in PLOS Genetics! Your manuscript is now with our production department and you will be notified of the publication date in due course.

With kind regards,

Amy Kiss

PLOS Genetics

On behalf of:
